# Fast Adversarial CNN-based Perturbation Attack on No-Reference Image Quality Metrics

**Ekaterina Shumitskaya**[1], **Anastasia Antsiferova**[2,3] **& Dmitriy Vatolin**[1,2,3]

[1] Lomonosov Moscow State University, Moscow, Russian Federation
[2] ISP RAS Research Center for Trusted Artificial Intelligence, Moscow, Russian Federation
[3] MSU Institute for Artificial Intelligence, Moscow, Russian Federation
`{ekaterina.shumitskaya, aantsiferova, dmitriy}@graphics.cs.msu.ru`

## Abstract

Modern neural-network-based no-reference image- and video-quality metrics exhibit performance as high as full-reference metrics. These metrics are widely used to improve visual quality in computer vision methods and compare video processing methods. However, these metrics are not stable to traditional adversarial attacks, which can cause incorrect results. Our goal is to investigate the boundaries of no-reference metrics applicability, and in this paper, we propose a fast adversarial perturbation attack on no-reference quality metrics. The proposed attack (FACPA) can be exploited as a preprocessing step in real-time video processing and compression algorithms. This research can yield insights to further aid in designing of stable neural-network-based no-reference quality metrics.

## 1 Introduction

State-of-the-art no-reference (NR) metrics commonly use neural-network-based approaches and deliver more accuracy than traditional ones. Quality-metric attacks lead such metrics to increase the quality score without improving visual quality. The developers of image- or video-processing and compression algorithms can exploit that vulnerability. For example, the developers of libaom (Deng et al., 2020) exploited the vulnerability of VMAF (VMAF: Perceptual video quality assessment based on multi-method fusion) by implementing the VMAF-oriented tuning option in the encoder. Our research aims to investigate the possibility of injecting quality-metric attacks into real-time algorithms, which can cause incorrect results in benchmarks. In this paper, we propose a Fast Adversarial CNN-based Perturbation Attack (FACPA) on NR quality metrics which is much faster than previous iterative attack methods and more effective than universal perturbation methods. Using this method, researchers can estimate the potential cheating gain of any NR metric used in benchmarks and video processing algorithms.

## 2 Related Work

Existing attack methods mainly focus on attacking image classification models (Szegedy et al., 2014), (Goodfellow et al., 2015), (Kurakin et al., 2017), (Carlini & Wagner, 2017), (Madry et al., 2018). Several studies on adversarial attacks related to NR quality metrics have been published. Sang et al. (2022) proposed an iterative algorithm based on adaptive distortions to attack quality metrics. This algorithm is based on the MI-FGSM (Dong et al., 2018) and controls distortions using the NR metric NIQE (Mittal et al., 2012). Korhonen & You (2022) proposed a method for the iterative attack of NR quality metrics that employs the Sobel filter to generate distortions on object edges since it allows to improve the visual quality of attacked images. Zhang et al. (2022) proposed an iterative attack to craft adversarial images using different full-reference metrics to control visual distortions. However, all these attacks are iterative and time-consuming, so their injection into real-time algorithms is unlikely. Shumitskaya et al. (2022) proposed to craft a universal adversarial perturbation to attack an NR image/video quality metric. This approach is much faster than iterative methods. However, this attack does not use information about the input and hence does not use the full potential of the attack.

## 3 PROPOSED METHOD AND RESULTS

Let $x_i$ be images from the training set of $N$ samples, $M$ — NR image/video quality metric to be attacked (e.g. MDTVSFA, Linearity, etc.). The proposed adversarial attack $f(x)$ can be formulated as follows:

$$\operatorname*{argmax}_{f}\{\frac{1}{N}\sum_{i=1}^{N}(M(x_i + f(x_i)) - M(x_i)))\}, \forall x_i : \|f(x_i)\|_\infty \leq \epsilon, \tag{1}$$

To find a function $f(x)$, we approximated it using U-Net encoder-decoder architecture with the tanh activation and scaling. We tried different approaches and chose U-Net architecture as it showed the best performance on preliminary tests. Constraint $\epsilon$ provided scaling. We used $\epsilon$ equal to $\frac{10}{255}$. For each target metric, we trained specific CNN weights. For CNNs training, we used 10,000 $256 \times 256$ images from the COCO dataset (Lin et al., 2014), defined the loss function as the target metric with the opposite sign and employed Adam optimizer (Kingma & Ba, 2014).

We compared the proposed attack with previous attacks on quality metrics using images from NIPS 2017: Adversarial Learning Development Set (Alex K, 2017). We attacked three quality metrics (Linearity (Li et al., 2020), VSFA (Li et al., 2019) and MDTVSFA (Li et al., 2021)) and compared the achieved increase of target metrics scores. For iterative attacks, we used a learning rate of 0.001 and attacked metrics using a different amount of iterations. Table 1 contains comparison results. FACPA is much faster than iterative methods (15 ms versus 110 – 5,000 ms) and more effective than the universal perturbation method. Figure 1 shows attack examples for the Linearity metric for compared methods.

Table 1: Percentage of metrics increase by proposed and other attacks and GPU calculation time.

| Attack method | It. | Linearity Gain ↑ | Linearity Time ↓ | VSFA Gain ↑ | VSFA Time ↓ | MDTVSFA Gain ↑ | MDTVSFA Time ↓ |
|---|---|---|---|---|---|---|---|
| Zhang et al. (2022) | 1 | 5.9% | 173 ms | 11.7% | 626 ms | 21.3% | 631 ms |
| | 10 | 26.1% | 1590 ms | 30.9% | 2660 ms | **47.9%** | 5980 ms |
| Korhonen & You (2022) | 1 | 5.6% | 111 ms | 11.1% | 483 ms | 20.4% | 505 ms |
| | 10 | 21.0% | 992 ms | **41.7%** | 4750 ms | 39.3% | 4040 ms |
| Sang et al. (2022) | 1 | 6.5% | 153 ms | 4.4% | 535 ms | 3.7% | 540 ms |
| | 3 | 18.6% | 358 ms | 32.6% | 1580 ms | 39.2% | 1580 ms |
| Shumitskaya et al. (2022) | - | 22.9% | **11 ms** | 22.0% | **11 ms** | 29.4% | **11 ms** |
| FACPA (ours) | - | **32.0%** | 15 ms | 30.6% | 15 ms | 40.3% | 15 ms |

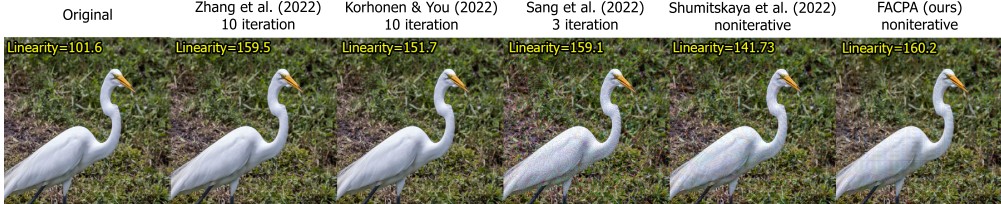

Figure 1: Attack examples for the Linearity quality metric.

## 4 CONCLUSION

In this paper, we proposed a fast CNN-based attack that increases the scores of NR quality metrics. The comparison with previous methods showed that our attack is much faster than iterative methods and more efficient than the universal adversarial perturbation. The speed of the proposed attack showed that it can be injected into video compression and other real-time algorithms. Therefore proposed attack can serve as an additional verification of metric reliability. Our code is publicly available at `https://github.com/katiashh/FACPA`.

URM STATEMENT

Authors Ekaterina Shumitskaya and Anastasia Antsiferova meets the URM criteria of ICLR 2023 Tiny Papers Track.

ACKNOWLEDGEMENTS

The work received support through a grant for research centers in the field of artificial intelligence (agreement identifier 000000D730321P5Q0002, dated November 2, 2021, no. 70-2021-00142 with the Ivannikov Institute for System Programming of the Russian Academy of Sciences).

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
