# OpenReview forum: "Fast Adversarial CNN-based Perturbation Attack on No-Reference Image- and Video-Quality Metrics"
_ICLR.cc/2023/TinyPapers — Submitted to Tiny Papers @ ICLR 2023_

### Official Review · Reviewer_nt4i · 2023-03-28

**Confidence:** 4

**Summary Of Contributions:**

 This paper studies fast adversarial attacks to NN-based quality metrics. The proposed attack is intended to be faster than iterative attacks and more effective than universal attacks.

**Rating:**

Clear, Correct, and Reproducible (CCR): a submission which meets the reviewing criteria

**Strengths And Weaknesses:**

Strengths:
- The paper presents an attack that is fast while achieving satisfactory effectiveness, for attacking quality metrics.

Weaknesses:
- The description of the proposed attack is not quite clear and detailed (how function f is obtained).


**Suggested Changes:**

Citation formats:
- "Adam optimizer Kingma & Ba (2014)." -> "Adam optimizer (Kingma & Ba, 2014)" (add parentheses, perhaps use \citep; many similar changes needed)

Description of the algorithm can be made more clear.

---

> ### Author Response · Authors · 2023-05-30
> **We have fixed the citation format. We also made the description of the proposed attacks more clear**
>
> Thank you for reviewing our paper! We have fixed the citation format. We also made the description of the proposed attacks more clear. Particularly, we emphasized that the function f defined in equation 1 was obtained by approximation using CNN. We trained a CNN with a loss function defined as the target metric M with the opposite sign. We also have published our code so that the readers can reproduce our results.

---

### Official Review · Reviewer_mohN · 2023-04-03

**Confidence:** 4

**Summary Of Contributions:**

The authors have proposed a fast adversarial perturbation attack (termed FACPA) on no-reference quality metrics, which may allow the quality metric to increase the quality scores without improving visual quality. Compared to previous methods, the proposed method is much faster, allowing better injection into real-time algorithms.

**Rating:**

Great Start (GS): a submission which meets some of the reviewing criteria but has room for improvement

**Strengths And Weaknesses:**

Strength:
1. Paper is clearly written. Motivation is also quite clear.
2. Comparison to other related methods seems adequate.


Weakness:
1. In the proposed attach equation, it is unclear to me what M (the target metric being attacked) actually is. The discussion seems missing.


**Suggested Changes:**

More discussion is needed on the target metric that is being attacked (M) in the proposed algorithm, with details.

---

> ### Author Response · Authors · 2023-05-30
> **We have added more details to the description and made it more clear**
>
> Thank you for reviewing our paper! We have added more details to the description and made it more clear, we also published our code so that the readers can reproduce our results. The target metric M is any differentiable NR image/video quality metric (for example, Linearity, VSFA, or MDTVSFA) to be attacked. The proposed attack is based on a white-box approach, which means that for each target metric, we train a specific CNN model for adversarial attacks.

---

### Meta-Review · Area_Chair_r6To · 2023-04-04

**Recommendation:** Invite to archive
**Confidence:** 4

**Metareview:**

The motivation is clear and the proposed method seems to be effective. But some essential details in the method are not quite clear and detailed.


**Summary:**

This paper studies adversarial attacks to image quality metrics. The proposed method is intended to be faster than existing iterative attacks.

**Reason For Not Giving A Higher Recommendation:**

Some essential details in the method are not quite clear and detailed.


**Reason For Not Giving A Lower Recommendation:**

The motivation is clear and the proposed method seems to be effective.

---

### Decision · Program_Chairs · 2023-04-08

Invite to archive